# Mapping the Genomic Regions Controlling Germination Rate and Early Seedling Growth Parameters in Rice

**DOI:** 10.3390/genes14040902

**Published:** 2023-04-12

**Authors:** Shakti Prakash Mohanty, Deepak Kumar Nayak, Priyadarsini Sanghamitra, Saumya Ranjan Barik, Elssa Pandit, Abhisarika Behera, Dipti Ranjan Pani, Shibani Mohapatra, Reshmi Raj K. R., Kartik Chandra Pradhan, Chita Ranjan Sahoo, Mihir Ranjan Mohanty, Chinmayee Behera, Alok Kumar Panda, Binod Kumar Jena, Lambodar Behera, Prasanta K. Dash, Sharat Kumar Pradhan

**Affiliations:** 1ICAR-National Rice Research Institute, Cuttack 753006, India; 2Department of Biosciences and Biotechnology, Fakir Mohan University, Balasore 756020, India; 3ICAR-National Bureau of Plant Genetic Resources, Base Center, Cuttack 753006, India; 4Environmental Science Laboratory, School of Applied Sciences, KIIT Deemed to be University, Bhubaneswar 751024, India; 5College of Agriculture, Odisha University of Agriculture & Technology, Bhubaneswar 751003, India; 6Directorate of Research, Odisha University of Agriculture & Technology, Bhubaneswar 751003, India; 7Regional Research and Technology Transfer Station (RRTTS), Odisha University of Agriculture & Technology, Jeypore 764001, India; 8Department of Genetics and Plant Breeding, Institute of Agricultural Sciences, SOA University, Bhubaneswar 753001, India; 9Krishi Vigyan Kendra, Odisha University of Agriculture & Technology, Rayagada 765022, India; 10ICAR-National Institute for Plant Biotechnology, Pusa, New Delhi 110012, India; 11Indian Council of Agricultural Research, Krishi Bhavan, New Delhi 110001, India

**Keywords:** absolute growth rate, association mapping, germination rate, *indica* rice, relative growth rate, relative shoot growth

## Abstract

Seed vigor is the key performance parameter of good quality seed. A panel was prepared by shortlisting genotypes from all the phenotypic groups representing seedling growth parameters from a total of 278 germplasm lines. A wide variation was observed for the traits in the population. The panel was classified into four genetic structure groups. Fixation indices indicated the existence of linkage disequilibrium in the population. A moderate to high level of diversity parameters was assessed using 143 SSR markers. Principal component, coordinate, neighbor-joining tree and cluster analyses showed subpopulations with a fair degree of correspondence with the growth parameters. Marker–trait association analysis detected eight novel QTLs, namely *qAGR4.1*, *qAGR6.1*, *qAGR6.2* and *qAGR8.1* for absolute growth rate (AGR); *qRSG6.1*, *qRSG7.1* and *qRSG8.1* for relative shoot growth (RSG); and *qRGR11.1* for relative growth rate (RGR), as analyzed by GLM and MLM. The reported QTL for germination rate (GR), *qGR4-1*, was validated in this population. Additionally, QTLs present on chromosome 6 controlling RSG and AGR at 221 cM and RSG and AGR on chromosome 8 at 27 cM were detected as genetic hotspots for the parameters. The QTLs identified in the study will be useful for improvement of the seed vigor trait in rice.

## 1. Introduction

The yield potential of a variety is realized by using quality seeds along with the recommended practices, which are important inputs for rice production. The future demand for the staple food rice is increasing due to a worldwide increase in population size. We need an additional production of about 1–2 MT of rice per year to fulfill our rice requirement by 2030 [1]. However, the higher production need beyond 2030 must be produced from less land, using less labor, water and chemicals, with a constant battle against new strains of pathogen and pests and under the adverse effects of the climate change [2,3,4]. Rice is life for the majority of the Indian population. Seed vigor is an important trait of a good quality seed that ensures germination, seedling growth and establishment of seedlings in the field that can withstand adverse climatic conditions [5]. Additionally, seed vigor is very important for direct-seeded rice since it enhances crop establishment [6,7] and increases weed competitiveness [8,9]. Seed vigor improvement is a breeding challenge in rice research [7,10]. Therefore, transfer of seedling vigor and its related parameters into high-yielding varieties of rainfed rice is very important.

Seed vigor in rice is a complex trait and controlled by many genes and QTLs associated with various morphological, physiological and biochemical traits [6,9,10]. Genetics of the seedling growth parameters are controlled by many quantitative trait loci (QTLs) and hence complex in nature. It is difficult to analyze the contribution of genetic factors by conventional genetic analysis due to the quantitative nature of inheritance of these traits. QTL analysis has become a powerful tool and successful approach for analyzing the inheritance of these complex traits during the last few decades. QTLs have been reported to be associated with many physiological growth traits, including dry weight of seedling, root length, shoot length, root activity, radicle length, coleoptile length, mesocotyl length, germination rate, germination potential, time for 50% germination and germination index. QTLs controlling the germination rate were reported in various mapping studies [11,12,13,14,15]. To highlight two major QTLs, *qEPD1* modulates plant growth both at the initial and heading stages, whereas *qEPD2* regulates early seedling plant height [16]. QTLs controlling shoot length were reported by earlier researchers [17,18,19,20]. Additionally, slow germination speed and seedling growth were reported to occur following disruption of *OsIPMS1,* the isopropylmalate synthase gene [21].

The results published on genetic analyses of seedling growth parameters were mostly derived from populations with a cross of two parents, while very few were obtained by association mapping. Genetic analysis using association mapping of growth parameters available in linkage disequilibrium (LD) population is a very successful approach to locating the chromosomal region responsible for a trait. QTL identification of multiple complex loci is possible with a high resolution and simultaneous testing when using this mapping approach that exploits the existing natural variation [22,23,24]. Genetic analysis using simple sequence repeat markers is useful for identifying a hyper variable, which is co-dominant, multi-allelic, robust and chromosome-specific in nature. These markers are used to assess genetic diversity and genetic structure in rice and several other crops and have greatly facilitated mapping of the complex traits [25,26,27,28]. The existence of genetic diversity in a panel for a trait is useful for detecting the marker–trait association, and the structure of the population provides an opportunity to develop a panel population in which unequal relatedness within the population or spurious association is rectified [2,29]. The population genetic structure (Q) and relative kinship (K) value are helpful to reconstitute the genotypes present in the panel population for mapping analyses [10,30,31]. Detection of QTLs and genes analyzed by GLM and MLM approaches provides high-quality mapping results for complex traits. However, there have been few genetic analyses of the growth parameters related to the early seedling stage utilizing a variable natural population in rice.

In this investigation, association mapping for the germination rate and early seedling growth parameters was performed in a representative panel population developed from 278 germplasm lines through phenotyping of four traits, namely germination rate, relative growth rate (RGR), AGR and MGR, and by utilizing 143 SSR markers. The study detected the population genetic structure, diversity and candidate genes/QTLs involved in the germination rate, including seedling growth parameters, in rice.

## 2. Material and Methods

### 2.1. Seed Material

The seeds of 278 diverse germplasm lines were obtained from Gene Bank, ICAR-NRRI, Cuttack, collected from the states of Assam, MP, Kerala, Odisha and Manipur of India. All germplasm lines were obtained from the Gene Bank and grown during the wet season of 2019 (Appendix A). The harvested seeds were stored for three months to overcome the seed dormancy of the freshly harvested seeds and then used for estimation of germination rate and early seedling growth parameters [32]. A panel population was developed from the phenotyping results of two replications conducted in the seed materials of the wet season in 2019. The panel population (124) was constituted and raised during the wet season in 2020 as the mapping population for the study on germination rate and early seedling growth parameters in rice (Table 1).

### 2.2. Phenotyping of Germination Rate and Early Seedling Growth Parameters in the Germplasm Lines

Seed physiological characteristics such as rate of shoot growth (RSG), absolute growth rate (AGR), mean germination rate (MGR) and relative growth rate (RGR) were calculated for the association study. A total of 50 seeds of each line in three replications were used for germination following the top of paper method by incubating at 30 °C. The estimate for RSG was performed by measuring the shoot length per day on the 7th and 10th days of germination and expressed in cm day^−1^. AGR is the rate of change of seedling length per day, whichwas calculated as per the procedure of Reford [33]. Seedling relative growth rate is the incremental accumulation of dry mass per unit of existing dry mass, which was estimated following the procedure Fisher [34]. Germination time is the time for the seed to germinate. The value of mean germination rate (MGR) is the velocity of germination. It is the inverse of mean germination time (MGT). MGR is expresses as day^−1^ and ranges from 0 to 1. MGR and MGT were calculated according to the formula of Ellis and Roberts [35], MGT (day) = ΣnD/N, where n is the number of seeds germinated on each day, D is number of days counted from the beginning of germination and N is the total number of seeds germinated. MGR (day^−1^) = 1/MGT. For estimation of the seedling growth parameters, five seedlings from each replication were recorded and the mean value of each germplasm line was estimated.

Analysis of variance (ANOVA) of the four traits was conducted using the software Cropstat 7.0. The other statistical parameters, namely range, mean and coefficient of variation (CV %), were computed using this software. The relationships among the four growth parameters were determined using Pearson’s correlation coefficient values based on the average values of the 124 germplasm lines, and a correlation matrix heat map was prepared. The genotypes were grouped into very high, high, medium and low value-carrying germplasm lines based on the seedling growth parameters for this association study.

### 2.3. Genomic DNA Isolation, PCR Analysis and Marker Selection

Rice seedlings of 15 days old were used for genomic DNA extraction by adopting the CTAB method [36]. A total of 143 SSR (simple sequence repeat) markers available in the public domain were used for the study (Appendix A). The DNA fragments isolated were quantified by resolving the fragments through agarose gel electrophoresis. PCR amplification was carried out by using the 143 selected SSR markers, which covered all the chromosomes and positions for identifying polymorphic loci and estimating the diversity of the 124 rice genotypes (Table 1). PCR reaction was performed by adopting the standard protocol of a denaturation step (4 min, 95 °C), followed by 35 cycles of denaturation (30 s, 94 °C) and annealing/extension (30 s, 55 °C), extension (1.5 min, 72 °C), final extension (10 min, 72 °C) and storage at 4 °C (infinity). Electrophoresis of the PCR products was performed using agarose gel (2.5%) containing 0.80 g mL^−1^ ethidium bromide. The sizes of the amplicons were estimated by comparing the known 50 bp DNA ladder in the electrophoresis. An electric current of 2.5 Vcm^−1^ was connected to run the gel for 4 h, and a photograph was captured in a Gel Documentation System (SynGene). The procedures adopted in earlier publications were followed in this isolation and electrophoresis work [37,38,39].

### 2.4. Molecular Data Analysis

Scoring of genotyping data was carried out for the absence or presence of amplified products on the basis of primer–genotype combination. The data in binary form wereprepared as discrete variables for the result data of the marker–genotype entries. The diversity parameters, namely major allele frequency (A), polymorphic information content (PIC), number of alleles (N), gene diversity (GD) and observed heterozygosity (H), were determined by using the software Power Marker Ver3.25′ for each SSR locus [40]. The population structure was generated by the software STRUCTURE 2.3.6 [41]. The ideal number of groups (K) was obtained by analysis with the software, which was run with K that varied from 1 to 10 and 10 iterations for each K value. The burn-in period for the high throughput was at 150,000, followed by 150,000 Markov Chain Monte Carlo (MCMC) replications adopted in the run. The value of ΔK of a subpopulation was obtained from the Evanno table, and the peak value was used. The peak value of L(K) was taken from the number of subpopulations. The most probable value of K was ΔK, which is a second-order change of the log probability value for the number of clusters detected by STRUCTURE [42]. The ΔK-value was obtained from the structure harvester, which is a function of K giving a clear peak for the optimal K-value [43]. The NEI coefficient dissimilarity index using an unweighted neighbor-joining unrooted tree with a bootstrap value of 1000 and the principal coordinate analysis for all the germplasm lines were performed by using DARwin5 software [44]. The F_IT_, F_IS_, F_ST_ for the presence of molecular variance across the population and within the population, including between the subpopulations, were estimated. Analysis of molecular variance (AMOVA) was performed using the GenAlEx 6.5 software [45].

The “TASSEL 5.0” software was used to detect the marker–trait associations for the four growth parameters. GLM and MLM were adopted for the association study of molecular markers and the growth parameters [46]. A significant *p*-value and *r*^2^ value were considered for markers with trait associations detected. The marker–trait associations were also checked by the quantile-quantile curve derived by the TASSEL software. An LD decay graph was generated by plotting the measured *r*^2^ value of a marker pair with the distance between the marker pair. The adjusted *p*-values (*q*-values) of the false discovery rate (FDR) were computed to check the accuracy of the marker–trait association using R software as described in previous publications [23,28].

## 3. Results

### 3.1. Phenotyping for Germination Rate and Early Seedling Growth Parameters in the Target Population

The mean values of 278 germplasm lines for germination rate and three seedling growth parameters were estimated during the wet season in 2019 (Appendix A). The germination rate and early seedling growth parameters such as rate of shoot growth, relative growth rate and absolute growth rate were estimated from the original population. Significant differences in the mean estimates of the four traits were noted among the germplasm lines. The frequency distributions of the 278 germplasm lines based on the mean phenotypic values were broadly classified into 5 groups for each parameter (Figure 1). The genotypes were distributed into different groups and further categorized into subpopulations (Figure 1). The panel population with 124 germplasm lines was prepared from the original 278 germplasm lines by shortlisting the lines from all the groups of each of the 4 studied traits (Table 1; Figure 2). The studied traits in the panel population also exhibited significant variation among the genotypes (Table 1). The estimates of RSG present in the panel showed very high values in the germplasm lines AC. 9030, AC. 9035, AC. 9038, Kapanthi and Pk-21 (Figure 2). In addition, a very high relative growth rate was recorded from the germplasm lines AC. 5993 and Kanakchampa (Table 1). The germplasm lines AC. 43660, AC. 43669 and AC. 43675 showed a high value of >3.0 for AGR (Table 1). In addition, the germplasm lines with a high mean germination rate were estimated from the germplasm lines AC. 43663, AC. 44603, AC. 44598, AC. 44592, AC. 44646, AC. 44604, AC. 44597, AC. 44638, AC. 44595, AC. 44588, AC. 44591 and AC. 44594. Landraces with values for all the four traits were observed in AC. 9006, AC. 9021, AC. 9028, AC. 9030, AC. 9035, AC. 9038, AC. 9043, AC. 9044, AC. 9063 and AC. 9058 (Table 1).

### 3.2. Principal Component and Association Analyses

The first two PCs (principal components) were used to generate a genotype-by-trait biplot diagram of the 4 studied physiological growth parameters estimated from the panel containing 124 germplasm lines (Figure 3A). Variations of 56.07% and 25.47% were exhibited by the first and second principal components, respectively. The scattering pattern revealed that the genotypes containing high estimates of growth parameters were placed in the 1st quadrant, which formed a cluster and accommodated 23 genotypes. Germplasm lines containing higher value for the MGR and growth parameters are depicted in a circle in the figure (Figure 3A). The top right (1st quadrant) and bottom left (3rd quadrant) accommodated 65 germplasm lines, forming the biggest group. The majority of the germplasm lines are of a moderate type for the early seedling growth parameters and germination rate with inclusion of a few good types. The 2nd (bottom right) quadrant kept most of the germplasm lines containing high RSG and MGR (Figure 3A).

The associations among the four physiological traits revealed significant positive correlations of RSG with AGR (0.447**) and MGR (0.551**). However, significant negative correlations were noted for RGR with RSG (−0.337**) and MGR (−0.402**) (Figure 3B).

### 3.3. Genetic Diversity Parameters’ Analysis

The representative population comprising 124 germplasm lines shortlisted from the original population showed wide variation for the four studied physiological growth parameters based on the analysis using 143 SSR markers. Gene diversity and other diversity-related parameters estimated for the marker loci are presented in Table 2. A total of 522 markers alleles were detected, showing 3.65 alleles per locus as an average value. The marker loci varied from two to seven alleles per locus, and the highest numbers of alleles were obtained by the marker RM493for the growth parameters. The estimated mean major allele frequency for the growth parameters associated with the markers was 0.582. The major allele frequency ranged from 0.274 (RM493) to 0.976 (RM14960) (Table 2). The variation for PIC value ranged from 0.133 (RM6054) to 0.792 (RM495) with an average value of 0.474. The estimated mean Ho (heterozygosity) in the panel population was 0.109, which ranged from 0.00 to 0.963. A zero heterozygosity value was estimated for 19 markers namely RM328, RM1812, RM6947, RM14978, RM22034, RM258, RM1347, RM3423, RM405, RM421, RM6091, RM209, RM245, RM3351, RM471, RM461, RM8007518, RM274 and RM452. The range for gene diversity (He) was from 0.150 (RM22034) to 0.817 (RM493), showing an average value of 0.530 for the panel.

### 3.4. Population Genetic Structure Analysis

The STRUCTURE 2.3.6 software was used to assess the genetic structure in the representative population by adopting probable subpopulations (K) and selecting a higher delta K-value for the studied physiological growth parameters. This grouped the germplasm lines into 2 subpopulations showing a high ∆K peak value of 349.8 at K = 2 (Appendix A). Subpopulation 1 showed an inferred ancestry value of 0.728, while 0.272 was obtained for subpopulation 2. Similarity was noted among the members of each subpopulation, but many deviations were observed in the inclusion of the germplasm lines for the studied growth parameters. Therefore, the next highest peak of the ∆K peak was taken, and the population was classified into four subpopulations, which showed better correspondence with the studied parameters and structure group compared to the two subpopulations at K = 2 (Figure 4A,B). The proportions of genotypes based on the inferred clusters were 0.184, 0.203, 0.512 and 0.1 for the subpopulations 1, 2, 3 and 4, respectively. The fixation index (Fst) values obtained by the software were 0.405, 0.351, 0.146 and 0.398 for the subpopulations 1, 2, 3 and 4, respectively. Cluster heterozygosity or expected mean distances were 0.297, 0.361, 0.399 and 0.368 for the subpopulations 1, 2, 3 and 4, respectively. The germplasm lines were grouped into a subpopulation based on an ≥80% ancestry value (Table 1; Figure 4).

The four seedling growth parameters exhibited a fair degree of correspondence among the structure subpopulation members present in the population and the studied trait at K = 4 compared to K = 2. The majority of the germplasm lines in subpopulation 1 showed the presence of a moderate value of growth parameters. Subpopulation 2 mainly accommodated genotypes showing high values, particularly for RSG, AGR and MGR traits. Subpopulation 3 accommodated germplasm lines carrying high values for one or more traits. Subpopulation 4 showed the presence of poor to moderate growth parameters carrying genotypes. The population also showed a low α value (α = 0.0683) estimated by the structure software at K = 4. The mean α-value had a positively leptokurtic distribution, while Fst_1_, Fst_2_, Fst_3_ and Fst_4_ showed almost symmetrical distributions with clear differentiation in the groups on the basis of distribution among the estimated Fst values (Appendix A).

### 3.5. Analysis of Molecular Variance (AMOVA) and LD Decay Plot

Plants presenting populations related with each other based on the studied growth parameters in the representative population were clustered together and different clusters formed different subpopulations. The estimated AMOVA showed the existence of genetic variations within and between the representative population at K = 4 (Table 3). The estimated genetic differentiation within and between the subpopulation at K = 4 was found to be 67% among individuals, 20% within individuals and 14% among the populations in the panel population. Wright’s F statistics estimates calculated for the four traits showed deviation from Hardy–Weinberg’s prediction. The parameters such as uniformity of individual within a subpopulation (F_IS_) and individual within the total population (F_IT_) were computed to determine the differentiation in the population. The F_IT_ and F_IS_ values based on the genotyping of 143 marker loci were 0.804 and 0.773, whereas F_ST_ was 0.138 among the subpopulations. The population differentiation is measured by the F_ST_ values or the subpopulations within the total population. Each subpopulation showed different F_ST_ values, and the distribution pattern of the genotypes also showed clear cut differentiation of the population into three subpopulations (Appendix A).

The association of markers with different traits was successfully used to map the traits through a disequilibrium study. The continuance of LD decay is an important factor for obtaining the disequilibrium in the population. The LD decay rate indicates the markers associated with the growth parameters that will be useful in the discovery of allelic variants or new genes regulating these studied traits. The syntenic *r*^2^ value was utilized to draw the LD decay plot of the population against the physical distance of the two markers in million base pairs (Figure 5A). The linked markers showed a decrease in *r*^2^ for an increase in the linkage distance. It was noted that the LD decay declined sharply for the linked markers at 1–1.5 mega base pairs and then the decay was gradual, with a very slow decay rate noted. Therefore, it is clear from the graph that LD decay is continuing for the four growth parameters in the population. The germplasm lines showing the admixture type may have originated in the evolution due to the existence of LD decay of these four traits. A similar trend is also observed in the marker R^2^, along with the marker ‘F’ versus marker ‘P’ plot (Figure 5B,C). This study indicated the strength of the marker–trait association with the associated markers for the studied traits.

### 3.6. Relatedness among the Germplasm Lines through Principal Coordinates and Cluster Analyses

The principal coordinate analysis (PCoA) for the two dimension diagrams is drawn on the basis of the genotyping results obtained by using the 143 SSR markers that grouped the genotypes for genetic relatedness among the germplasm lines (Figure 6). The inertia for component 1 and component 2 were 11.6% and 7.27% of total inertia, respectively. The germplasm lines were placed on the four quadrants at different spots, which formed four major groups (Figure 7). A total of 13, 13, 18 and 80 germplasm lines were allocated into the 1st, 2nd, 3rd and 4th quadrant, respectively. The germplasm lines of each subpopulation are clustered in different quadrants. The 4th quadrant showed a single major group accommodating 80 germplasm lines. The 2nd major group present in the 1st and 2nd quadrants accommodated 23 germplasm lines. Genotypes from subpopulation 1 and subpopulation 2 are placed together in this group. The majority of the germplasm lines of this group are good for seedling growth parameters. The admix genotypes are depicted in brick color (Figure 6). The majority of the members of subpopulation 3 are in quadrant 4 and depicted in green color. Eight subpopulation 4 members along with seven admix types, two subpopulation 3 members and one subpopulation 1 member are allotted in the 3rd quadrant.

The germplasms UPGMA tree constructed based on the results of genotyping of 143 SSR markers in the representative population grouped the panel’s genotypes into 4 groups as in the case of the PCoA plot. The colors of the four subpopulations depicted in the tree are blue for SP1; pink for SP2; green for SP3; violet for SP4; and red for admix type (Figure 7A). The unweighted neighbor-joining tree classified the panel population into four different subpopulations including the admix types. The tree discriminated the germplasm lines into different clusters on the basis of the genotyping results using 143 SSR markers that corresponded with the germination rate and 3 early seedling growth parameters. The cluster accommodating subpopulation 3 was differentiated from SP2 by the presence of lines carrying high values for MGR and RSG, while moderate to high genotypes were seen for all four parameters in subpopulation 1. The admix types of germplasm lines in the population shown in the neighbor-joining tree are in red (Figure 7A). A phylogenetic tree was also constructed using the unrooted tree. This tree lacks a common ancestor or node. Here, the distances of each germplasm line are depicted in the diagram (Figure 7B). Both the trees are useful for finding the relationship between germplasm lines irrespective of the evolutionary time.

The panel population is broadly grouped into two clusters based on the phenotypic values of the four growth parameters of the germplasm lines. Again, this cluster is divided into two clusters. Finally, 10 different subgroups were seen in the dendrogram based on the values of the four seedling growth parameters (Figure 8). Subcluster I was the largest group, which accommodated 48 genotypes, while subcluster III was the smallest one, with only 4 germplasm lines. Cluster I mainly accommodated the germplasm lines with high AGR and RSG. Subclusters III and IV of cluster II separated out the genotypes carrying high estimates of three or four parameters. Subclusters I and II of cluster II accommodated genotypes mainly with high RSG and MGR. Admix genotypes were observed in cluster I, while no admix was seen in cluster II.

### 3.7. Association of Marker Alleles with Germination Rate and Early Seedling Growth Parameters in Rice

Marker–trait associations for four physiological traits were computed by the TASSEL 5 software using both GLM (generalized linear model) and MLM (mixed linear model)/K+Q models. The estimates of the associations were subjected to filtration at <1% error, i.e., 99% confidence (*p* < 0.01). Three growth parameters, namely RSG, RGR and MGR, showed significant associations with markers using both GLM and MLM analyses at *p* < 0.01. However, all four traits were significantly associated with markers analyzed by GLM and MLM separately. A total of 58 and 25 significant marker–trait associations were obtained when analyzed by GLM and MLM, respectively, at *p* < 0.01. The estimated marker R^2^ value when using GLM analysis varied from 0.027 to 0.109, while the value varied from 0.049 to 0.116 with MLM analysis (Appendix A). A total of nine significant marker–trait associations were obtained for RSG, RGR, AGR and MGR by both the models at >0.05 markers R^2^ and *p* < 0.01. AGR showed a significant association with four markers; RSG with three markers; and RGR and MGR with one marker each by both the models at *p* < 0.01 (Table 4). The Q-Q plot also confirmed these marker–trait associations for the seedling stage physiological parameter traits in rice (Figure 9).

High *r*^2^ values > 0.1 were detected from the marker–trait association of markers RM337 and RM494 for the trait RSG. The three markers associated with the parameter RSG, namely RM337, RM22034 and RM494, are present at 27, 56 and 221 cM positions on chromosomes 8, 7 and 6, respectively (Table 4). Five markers were significantly associated with AGR, as detected by both generalized linear model and mixed linear model analyses at *p* < 0.01 and *r*^2^ value > 0.05. The chromosomal regions governing the trait AGR were detected on chromosomes 8, 6 and 4. Among the four markers, RM16686 showed the highest marker R^2^ value of 0.076 analyzed by GLM and 0.095 by MLM. The strongly associated marker is located on chromosome 4 at the 300 cM position. RM337, RM7179 and RM494 were significantly associated with the trait at 27, 159 and 221 cM positions on chromosomes 8, 6 and 6, respectively. In addition, two markers, namely RM1812 and RM3735, were significantly associated with the growth parameters RGR and MGR at 44 and 80 cM positions on chromosomes 11 and 4, respectively. The Q-Q plot also confirmed these marker–trait associations for the four physiological growth parameters in rice (Figure 9).

The common markers were detected to be associated with more than one seedling stage growth parameter in rice. Two markers, RM494 and RM337, exhibited significant associations with two physiological growth parameters, namely RSG and AGR, by both the models at <1% error (Table 4). The markers are present at 221 and 27 cM positions on chromosomes 6 and 8, respectively.

## 4. Discussion

The genotypes present in the population were significantly different from each other for the four studied seedling growth parameters (Table 1). The studied physiological growth parameters at the seedling stage also showed significant correlation among themselves. The panel population showed higher genetic variations and higher correlation coefficients for the growth parameters, indicating the usefulness of the population for improvement of RSG, RGR, AGR and MGR in rice (Table 1; Figure 3B). Reports of usefulness of high variations for many traits in crop improvement programs were previously published [47,48,49,50]. Phenotypic variations for the 4 traits and the existence of diversity estimated based on the genotyping results using 143 markers confirmed the differentiations of the whole population into subpopulations (Table 2). Existence of more alleles and moderate to high PIC in the population indicated that markers are informative, which may be useful in seedling growth parameter improvement programs. The germplasms used in this study were collected from the states where rich rice genetic diversity exists. In the present experiment, germplasm lines from the Jeypur tract of Odisha were used, which is the secondary center of origin of rice. The germplasm lines AC. 9006, AC. 9021, AC. 9028, AC. 9030, AC. 9035, AC. 9038, AC. 9043, AC. 9044, AC. 9058 and AC. 9063 were high in the content of the four studied traits. These germplasm lines will be useful as potential donors for the seedling-stage growth parameters in breeding programs (Table 1). Therefore, it is expected that the identified donor lines from this population will be useful in improvement of seed vigor and its related traits in rice. The availability of genetic diversity in rice germplasms has been reported by many earlier researchers [51,52,53,54,55,56,57,58]. The presence of different Fst values and four structure groups supported the existence of various LD groups in the population. Existence of admix-type landraces along with a low α value in the population revealed that these traits were originated from a single source and formed many admix races with different ancestry values in the evolutionary process. Earlier workers have supported the correspondence of structure and growth parameters in rice [31,59,60,61,62]. Additionally, many publications on the correlation of phenotypes of various traits with structure subpopulations have been published [59,60,63].

All four physiological growth parameters were analyzed by both GLM and MLM approaches and found to be associated with nine SSR markers (Table 4). The markers found to be associated with traits at *p* < 0.01 with a low ‘*p*’ value and detected by both the models are considered as robust and very useful for breeding programs. The markers, namely RM337, RM22034, RM494, RM1812, RM7179, RM16686 and RM3735, will be useful for improvement of seedling growth parameters through molecular breeding for enhancing vigor in rice (Table 4). The quantile-quantile (Q-Q) plot also confirmed the detected associated markers with the growth parameters in rice (Figure 9). Marker–trait associations have been reported previously by many workers in rice [24,30,64].

QTLs for the germination rate in rice were reported in previous studies [11,12,13,14,15,65]. QTLs controlling the germination rate located on chromosome 4 were reported by Wang et al. [12] and Yang et al. [15]. Wang et al. reported QTL in the marker interval of RM252–RM317, which is at 102.2–117.5 cM position. As per the report of Yang et al. [15], the QTL is located from 27.3 MB to 28.4 MB. We detected the association for the trait on the chromosome at 80 cM, which is nearer to the QTL reported by Wang et al. [12]. In addition, we detected at a physical distance of 26.2 MB of the marker. Therefore, our detected QTL may be *qGP-4* as per Wang et al. [12] or *qGR4-1* by Yang et al. [15]. We detected four significant associations of markers with the trait absolute growth rate. The associations were at 300 cM position on chromosome 4; 159 and 221 cM on chromosome 6; and 27 cM position on chromosome 8. Bharamappanavara et al. [64] reported a QTL on chromosome 2 at 61.59–115.44 cM position and on chromosome 3 within 16.95 cM. However, no reports are available for the locations detected by us for QTLs controlling AGR in rice. These QTLs are designated as *qAGR4.1* on chromosome 4; *qAGR6.1* and *qAGR6.2* on chromosome 6; and *qAGR8.1* on chromosome 8.

Different reports on genetic analysis of growth parameters at the seedling stage were published [65,66]. The growth parameter relative shoot length showed a significant association with the markers RM337, RM22034 and RM494 detected by both GLM and MLM analyses. The bi-parental mapping study of Han et al. located the QTL on chromosome 1 [67]. The team [67] reported a QTL on chromosome 12 at 101–107 cM position. We detected the associations at 27, 56 and 221 cM positions on the chromosomes 8, 7 and 6, respectively. Abe et al. [18], Dang et al. [68] and Anandan et al. [22] also reported the locus controlling the trait but on different chromosomes. The QTLs detected by us are designated as *qRSG8.1, qRSG7.1* and *qRSG6.1* on the chromosomes 8, 7 and 6, respectively (Figure 9). Kato et al. [69] mapped the QTL for relative growth rate on chromosome 4 within the marker interval of RM8213–RM335 and the other QTL on chromosome 7 within RM8249–RM5120. The identified loci in this experiment were present on chromosome 11, which was not reported in earlier studies. Hence, the locus we detected was a new QTL, designated as *qRGR11.1*.

The QTLs controlling RSG and AGR, namely *qRSG6.1* with *qAGR6.2* on chromosome 6 at 221 cM position, were detected to be co-inherited. Similarly, the QTLs *qRSG8.1* and *qAGR8.1* on chromosome 8 at 27 cM position for both the traits were detected to be co-localized with each other. In earlier mapping studies, there were reports of co-localization of genes/QTLs controlling traits such as protein, Fe, Zn and antioxidant contents in grains and different growth parameters in rice [24,28,60,62]. It is very easy to improve the traits controlled by the QTL hotspots located on the chromosomes.

## 5. Conclusions

Seed quality and vigor can be improved by improving the seed germination rate and the early seedling growth parameters in rice. The seedling growth parameters such as germination rate, RSG, RGR and AGR showed wide variations in a panel of 124 genotypes that represented 278 genotypes. High values of the growth parameters were identified in the landraces such as AC. 9006, AC. 9021, AC. 9028, AC. 9030, AC. 9035, AC. 9038, AC. 9043, AC. 9044, AC. 9063 and AC. 9058. Based on the fixation indices values computed from the subpopulations, the presence of LD was confirmed in the panel population. Existence of moderate to high PIC values, gene diversity and related parameters were obtained in the population by genotyping with a set of 143 markers. The population in the panel was categorized into few subpopulations and subclusters, which had good correspondence with the members for the four studied parameters. A total of eight novel QTLs controlling the traits, namely *qAGR4.1, qAGR6.1, qAGR6.2* and *qAGR8.1* for AGR; *qRSG6.1*, *qRSG7.1* and *qRSG8.1* for RSG; and *qRGR11.1* for RGR, were detected from the mapping population. The reported QTL for germination rate, *qGR4-1,* was validated in this mapping population and will be useful in marker-assisted breeding. Additionally, QTLs present on chromosome 6 controlling RSG and AGR at 221 cM and on chromosome8 at 27 cM for RSG and AGR were detected as genetic hotspots for the parameters. The QTLs identified in the study will be useful for crop improvement programs for seed-vigor-related traits in rice.

## Figures and Tables

**Figure 1 genes-14-00902-f001:**
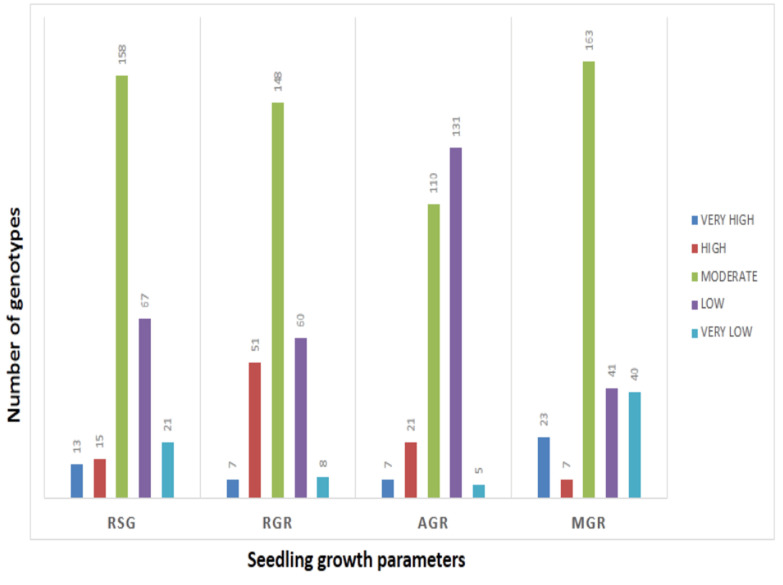
Frequency distribution of germplasm lines for each of the seedling stage physiological parameters for RSG, AGR, RGR and GR estimated from 278 germplasm lines.

**Figure 2 genes-14-00902-f002:**
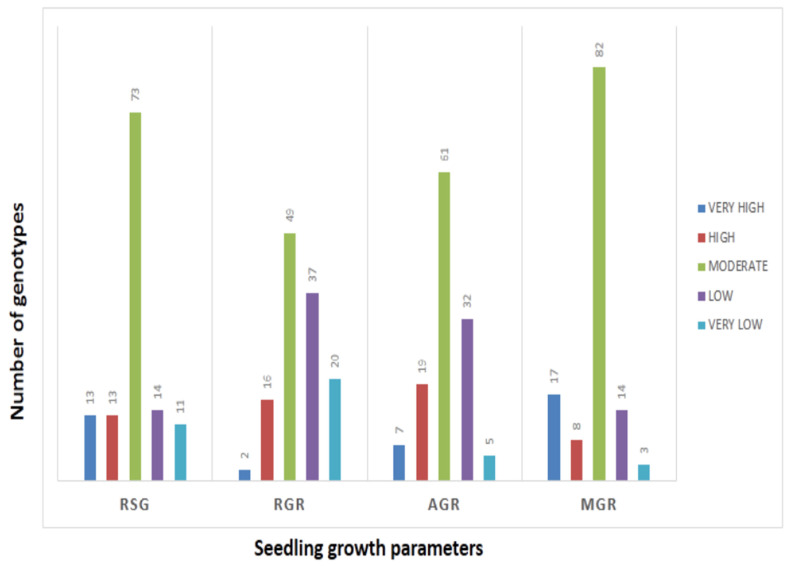
Frequency distribution of genotypes present in the panel population by histogram for RSG, AGR, MGR and RGR.

**Figure 3 genes-14-00902-f003:**
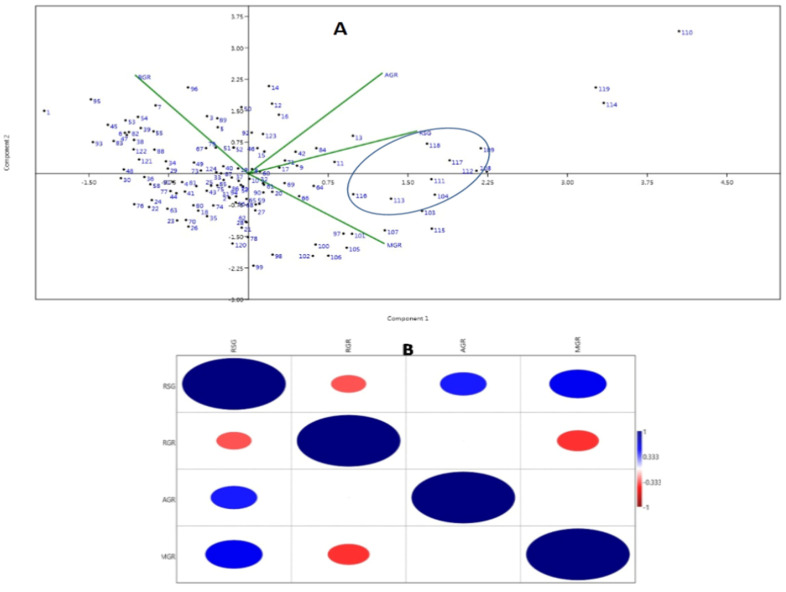
(**A**) Principal component analysis biplot diagram showing 124 landraces in two PCs for 4 seedling stage growth parameter. (**B**) Heat map showing Pearson’s correlation coefficients for the growth parameters. AGR: Absolute growth rate; MGR: mean germination rate; RGR: relative growth rate and RSG: rate of shoot growth. The numbers in the dot present in the diagram depict the serial numbers of the genotypes listed in Table 1. Significant correlations are colored either in red (negative correlation at 0.01 level) or blue (positive correlation at 0.01 level).

**Figure 4 genes-14-00902-f004:**
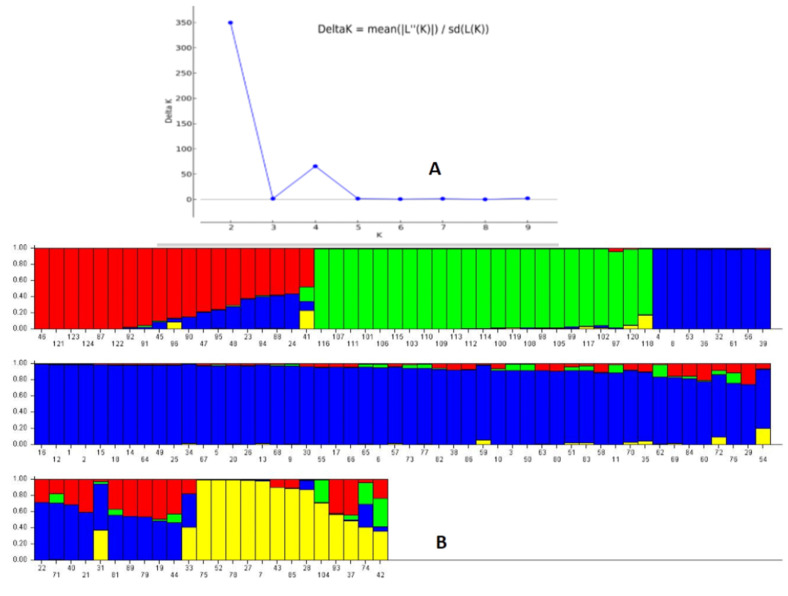
(**A**) Plot of ∆K value and change in the log probability of data between successive K values; (**B**) population genetic structure obtained for the 124 germplasm lines in the panel population based on membership probability at K = 4. The germplasm lines showing ≥80% membership values were assigned as members of the subgroup, while others were admixture type. The serial numbers of the genotypes present in the diagram are as per the germplasm serial numbers in Table 1.

**Figure 5 genes-14-00902-f005:**
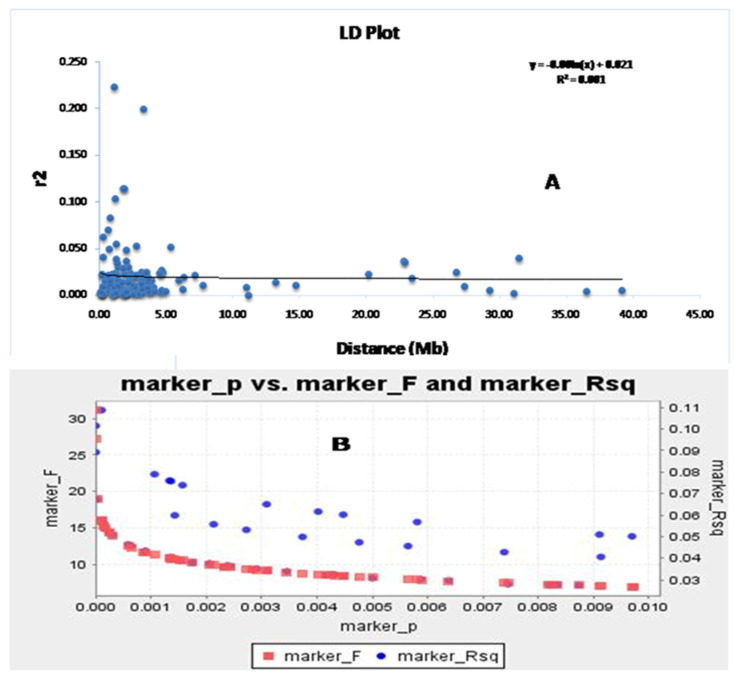
The marker ‘P’ versus marker ‘F’ and marker R^2^ detected using (**A**) the GLM approach and (**B**) the physical distance (Mb) between pairs of marker loci on chromosomes vs. a linkage disequilibrium decay (*r*^2^) curve plotted in rice. The decay started in million bp estimated by considering the 95th percentile of the distribution of *r*^2^ for all unlinked loci.

**Figure 6 genes-14-00902-f006:**
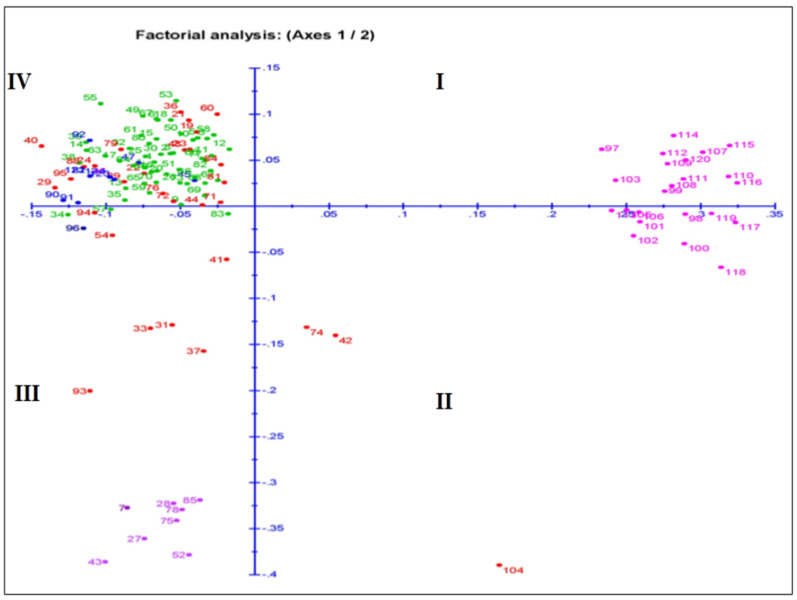
Plot for principal coordinate analysis of 124 genotypes present in the population for 4 growth parameters by genotyping with 143 microsatellite markers. The numbers in the dots indicate the serial of the landrace present in Table 1. The numbers are colored on the basis of sub-populations obtained from the structure analysis at K = 4 (SP1: blue; SP2: pink; SP3: green; SP4: violet; admix: red).

**Figure 7 genes-14-00902-f007:**
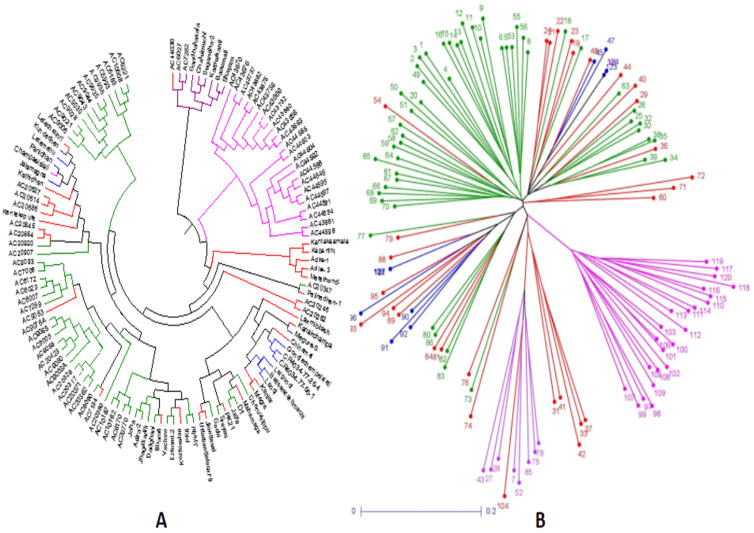
Trees constructed based on the genotyping results of 124 landraces using 143 SSR markers for depicting clustering patterns: (**A**) UPGMA unrooted tree; (**B**) neighbor-joining tree colored based on the subpopulations obtained from the structure analysis (SP1: blue; SP2: pink; SP3: green; SP4: violet; admix: red).

**Figure 8 genes-14-00902-f008:**
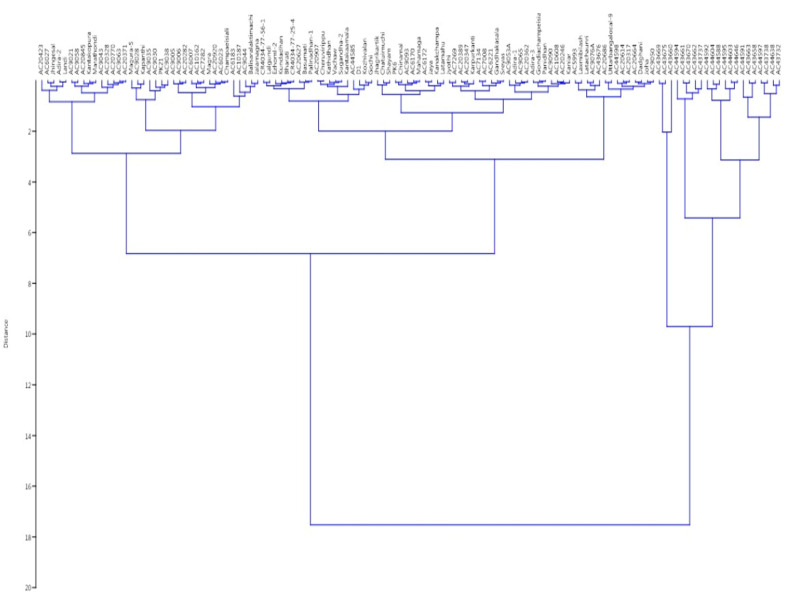
Wards’ clustering based on the estimates of 4 physiological traits for clustering of 124 germplasm lines.

**Figure 9 genes-14-00902-f009:**
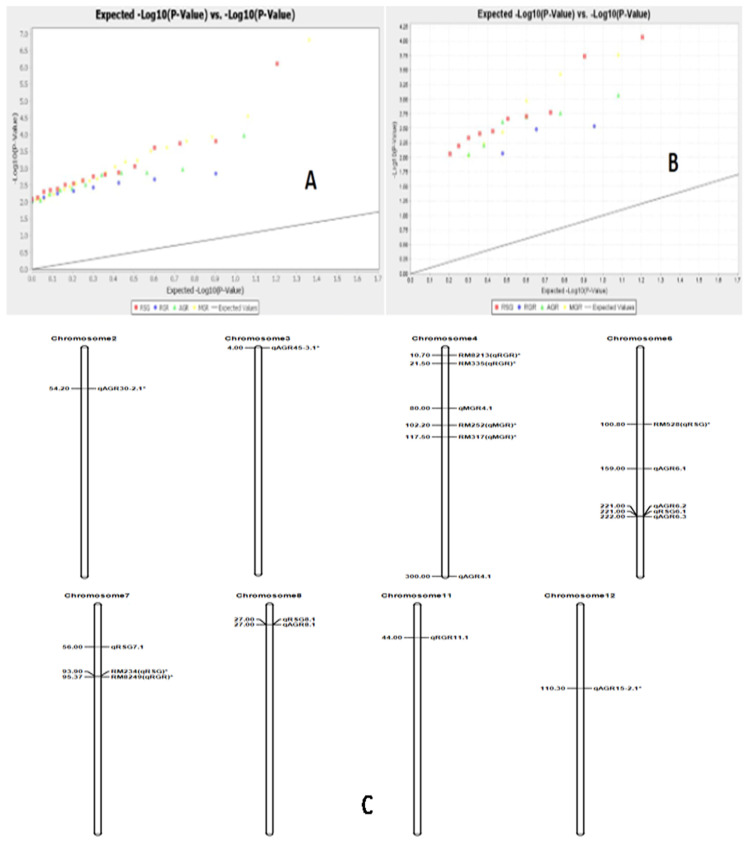
The Q-Q diagram constructed by analysis with the generalized linear model for the four growth parameters (**A**) *p* < 0.05, (**B**) *p* < 0.01) and(**C**) the positions of the QTLs on the chromosomes for RSG, RGR, AGR and MGR detected by association mapping in rice.

**Table 1 genes-14-00902-t001:** Mean estimates of RSG, RGR, ASR and MGR and genetic structure ancestry value at K = 4 in the panel population (n = 124).

Sl. No.	Accession No./ Vernacular Name of the Germplasm Line	Rate of Shoot Growth (RSG)	Relative Growth Rate (RGR)	Absolute Growth Rate (AGR)	Mean Germination Rate (MGR)	Inferred Ancestry Value at K = 4					Germplasm Lines with High Seedling Growth Parameters
						Q1	Q2	Q3	Q4	Group	
1	AC. 5993	0.317	0.347	0.577	0.182	0.007	0.004	0.985	0.004	SP3	RGR
2	AC. 6221	0.667	0.105	1.283	0.209	0.004	0.009	0.984	0.003	SP3	RGR
3	AC. 6183	0.520	0.186	2.137	0.210	0.003	0.081	0.913	0.003	SP3	RGR, AGR
4	AC. 6170	0.577	0.108	1.023	0.202	0.002	0.002	0.993	0.002	SP3	
5	AC. 6023	0.947	0.126	2.017	0.182	0.009	0.015	0.973	0.002	SP3	AGR
6	AC. 6172	0.440	0.261	1.040	0.213	0.007	0.042	0.949	0.002	SP3	
7	AC. 6027	1.017	0.241	1.570	0.224	0.002	0.002	0.013	0.982	SP4	RSG, MGR
8	AC. 6007	0.880	0.034	2.053	0.195	0.002	0.003	0.993	0.002	SP3	AGR
9	AC. 9006	1.227	0.120	2.063	0.284	0.005	0.018	0.964	0.012	SP3	RSG, AGR, MGR
10	AC. 9021	1.099	0.146	1.339	0.279	0.059	0.021	0.914	0.006	SP3	RSG, MGR
11	AC. 9028	1.373	0.114	2.297	0.305	0.008	0.105	0.884	0.004	SP3	RSG, AGR, MGR
12	AC. 9030	1.767	0.239	2.293	0.267	0.006	0.006	0.987	0.001	SP3	RSG, RGR, AGR, MGR
13	AC. 9035	1.800	0.113	2.717	0.283	0.01	0.004	0.968	0.019	SP3	RSG, AGR, MGR
14	AC. 9038	1.623	0.276	2.527	0.273	0.013	0.002	0.983	0.002	SP3	RSG, RGR, AGR, MGR
15	AC. 9043	1.363	0.221	1.643	0.309	0.012	0.002	0.984	0.002	SP3	RSG, RGR, MGR
16	AC. 9044	1.103	0.272	2.350	0.320	0.003	0.003	0.988	0.006	SP3	RSG, RGR, AGR, MGR
17	AC. 20920	1.120	0.039	2.073	0.219	0.037	0.004	0.954	0.006	SP3	RSG, AGR
18	AC. 20907	0.523	0.066	0.790	0.225	0.013	0.002	0.984	0.001	SP3	
19	AC. 20845	1.030	0.101	1.540	0.227	0.487	0.023	0.489	0.001	Admix	RSG
20	AC. 20770	1.207	0.049	1.493	0.243	0.015	0.004	0.973	0.008	SP3	RSG
21	AC. 20627	0.700	0.013	0.927	0.241	0.402	0.002	0.595	0.001	Admix	
22	AC. 20686	0.203	0.159	0.350	0.250	0.282	0.003	0.714	0.002	Admix	
23	AC. 20664	0.290	0.080	0.510	0.232	0.616	0.01	0.372	0.002	Admix	
24	AC. 20614	0.193	0.148	0.513	0.237	0.558	0.001	0.437	0.004	Admix	
25	Jhagrikartik	0.337	0.096	1.307	0.232	0.012	0.009	0.977	0.002	SP3	
26	Dadghani	0.340	0.096	0.450	0.259	0.021	0.006	0.969	0.004	SP3	MGR
27	Shayam	0.473	0.070	1.283	0.277	0.003	0.002	0.004	0.991	SP4	MGR
28	Basumati	0.663	0.066	0.953	0.277	0.006	0.005	0.116	0.873	SP4	MGR
29	Bharati	1.047	0.183	0.873	0.230	0.253	0.001	0.742	0.004	Admix	RSG, RGR
30	Joha	0.280	0.183	0.500	0.209	0.033	0.002	0.962	0.002	SP3	RGR
31	Adira-1	0.887	0.069	1.117	0.232	0.026	0.03	0.57	0.374	Admix	
32	Adira-2	0.890	0.065	1.643	0.242	0.005	0.002	0.991	0.002	SP3	
33	Adira-3	0.840	0.109	1.373	0.227	0.173	0.005	0.413	0.409	Admix	
34	PK6	0.620	0.157	1.083	0.204	0.002	0.004	0.977	0.017	SP3	RGR
35	Vachaw	0.413	0.038	0.877	0.221	0.099	0.002	0.855	0.044	SP3	
36	Kozhivalan	0.527	0.184	0.620	0.224	0.005	0.003	0.992	0.001	Admix	RGR
37	Marathondi	1.067	0.057	1.427	0.210	0.441	0.056	0.014	0.489	Admix	RSG, AGR
38	Ezhoml-2	0.677	0.257	0.920	0.252	0.073	0.001	0.924	0.002	SP3	RGR, MGR
39	Jyothi	0.737	0.251	1.160	0.269	0.009	0.001	0.989	0.002	SP3	RGR, MGR
40	Kantakopura	0.980	0.121	1.427	0.227	0.312	0.002	0.684	0.002	Admix	
41	Kantakaamala	0.370	0.082	1.010	0.206	0.478	0.178	0.114	0.23	Admix	
42	Kapanthi	1.513	0.066	2.190	0.224	0.234	0.35	0.055	0.361	Admix	AGR
43	Karpurkanti	0.707	0.068	1.110	0.208	0.093	0.001	0.001	0.905	SP4	
44	Kathidhan	0.447	0.089	0.867	0.204	0.423	0.112	0.461	0.005	Admix	
45	Kundadhan	0.777	0.246	0.933	0.173	0.897	0.013	0.088	0.003	SP1	RGR
46	Champaeisiali	0.947	0.104	2.060	0.224	0.996	0.001	0.002	0.001	SP2	AGR
47	Latamahu	0.323	0.221	1.153	0.191	0.779	0.009	0.208	0.003	SP1	RGR
48	Latachaunri	0.237	0.197	0.653	0.212	0.705	0.011	0.281	0.003	Admix	
49	AC. 10608	0.850	0.121	1.290	0.200	0.014	0.006	0.979	0.001	SP3	AGR
50	AC. 10187	0.667	0.157	2.583	0.206	0.005	0.079	0.912	0.003	SP3	RGR, AGR
51	AC. 10162	0.890	0.073	1.957	0.182	0.04	0.043	0.893	0.023	SP3	
52	AC. 7282	0.990	0.075	1.917	0.212	0.002	0.001	0.002	0.995	SP4	MGR
53	AC. 7269	0.640	0.244	1.210	0.184	0.004	0.003	0.993	0.001	SP3	RGR
54	AC. 7134	0.690	0.233	1.377	0.182	0.058	0.01	0.727	0.205	Admix	RGR
55	AC. 7008	0.700	0.191	1.360	0.182	0.042	0.002	0.955	0.001	SP3	RGR
56	AC. 9093	0.573	0.150	1.030	0.306	0.003	0.002	0.99	0.005	SP3	RGR, MGR
57	AC. 9090	0.797	0.147	1.273	0.288	0.032	0.006	0.945	0.018	SP3	MGR
58	AC. 9076A	0.133	0.195	0.677	0.252	0.102	0.011	0.886	0.002	SP3	MGR
59	AC. 9065	0.883	0.121	1.153	0.302	0.012	0.005	0.921	0.062	SP3	RGR, MGR
60	AC. 9063	1.190	0.126	1.557	0.264	0.202	0.014	0.783	0.001	Admix	RSG, MGR
61	AC. 9058	1.170	0.115	1.407	0.275	0.007	0.001	0.991	0.001	SP3	RSG, MGR
62	AC. 9053A	1.057	0.013	1.197	0.255	0.009	0.155	0.823	0.013	SP3	RSG, MGR
63	AC. 9050	0.257	0.151	0.473	0.261	0.079	0.003	0.909	0.009	SP3	RGR
64	AC. 9005	1.109	0.077	1.912	0.293	0.01	0.005	0.981	0.003	SP3	RSG, MGR
65	AC. 20389	0.707	0.109	1.220	0.280	0.003	0.034	0.951	0.011	SP3	MGR
66	AC. 20371	1.193	0.100	1.540	0.307	0.041	0.006	0.952	0.001	SP3	RSG, AGR, MGR
67	AC. 20423	0.590	0.142	1.680	0.213	0.018	0.005	0.976	0.001	SP3	AGR
68	AC. 20362	1.130	0.100	1.187	0.286	0.008	0.017	0.968	0.007	SP3	RSG, AGR, MGR
69	AC. 20328	1.120	0.100	1.660	0.281	0.155	0.012	0.816	0.018	SP3	RSG, AGR, MGR
70	AC.20317	0.150	0.146	0.497	0.288	0.078	0.004	0.884	0.035	SP3	MGR
71	AC. 20282	1.150	0.094	2.033	0.251	0.172	0.12	0.695	0.013	Admix	RSG, AGR, MGR
72	AC. 20246	0.893	0.123	1.297	0.266	0.084	0.047	0.775	0.094	Admix	MGR
73	AC. 20347	0.693	0.130	1.257	0.224	0.008	0.048	0.942	0.002	SP3	
74	Palinadhan-1	0.660	0.075	0.927	0.236	0.039	0.267	0.284	0.411	Admix	
75	Chatuimuchi	0.527	0.142	1.190	0.266	0.001	0.001	0.001	0.996	SP4	MGR
76	Uttarbangalocal-9	0.107	0.159	0.307	0.236	0.114	0.124	0.76	0.002	Admix	RGR
77	Gochi	0.447	0.159	0.697	0.246	0.006	0.048	0.937	0.009	SP3	RGR
78	Sugandha-2	0.433	0.029	0.890	0.273	0.002	0.001	0.003	0.993	SP4	MGR
79	Jhingesal	0.937	0.182	1.547	0.243	0.458	0.002	0.539	0.001	Admix	RGR
80	Cheruvirippu	0.503	0.084	0.803	0.228	0.088	0.004	0.906	0.002	SP3	
81	Mahamaga	0.617	0.100	1.060	0.218	0.37	0.071	0.557	0.002	Admix	
82	Jaya	0.507	0.273	1.033	0.223	0.057	0.009	0.933	0.002	SP3	RGR
83	D1	0.483	0.241	0.840	0.198	0.028	0.057	0.887	0.028	SP3	RGR
84	PK21	1.613	0.059	2.400	0.231	0.152	0.031	0.814	0.002	Admix	RSG, AGR
85	Gandhakasala	0.670	0.091	1.390	0.227	0.106	0.004	0.002	0.888	SP4	
86	Sreyas	0.707	0.064	1.373	0.220	0.071	0.003	0.923	0.004	SP3	
87	Gondiachampeisiali	0.910	0.084	1.423	0.222	0.995	0.001	0.002	0.001	SP1	
88	Chinamal	0.533	0.199	1.117	0.212	0.578	0.002	0.409	0.011	Admix	RGR
89	Magra	1.060	0.180	2.073	0.211	0.451	0.003	0.541	0.005	Admix	RSG
90	Landi	0.843	0.030	1.590	0.229	0.847	0.003	0.148	0.002	SP1	
91	Lalgundi	0.600	0.143	1.443	0.224	0.956	0.017	0.022	0.005	SP1	RGR
92	Balisaralaktimachi	0.977	0.141	2.170	0.227	0.976	0.007	0.014	0.003	SP1	
93	Laxmibilash	0.377	0.259	0.650	0.256	0.416	0.006	0.013	0.565	Admix	MGR
94	Kaniar	0.843	0.051	1.250	0.225	0.579	0.014	0.394	0.012	Admix	MGR
95	Kanakchampa	0.457	0.308	1.190	0.179	0.756	0.008	0.223	0.014	Admix	RGR
96	Magura-S	1.350	0.246	2.097	0.189	0.862	0.003	0.043	0.092	SP1	RGR, AGR
97	AC. 44603	2.050	0.044	1.100	0.333	0.036	0.945	0.018	0.002	SP2	RSG, MGR
98	AC. 44585	0.750	0.064	0.611	0.333	0.004	0.978	0.006	0.013	SP2	MGR
99	AC. 44598	0.200	0.056	0.500	0.333	0.003	0.968	0.019	0.01	SP2	MGR
100	AC. 44592	1.517	0.047	0.883	0.333	0.002	0.982	0.002	0.015	SP2	RSG, MGR
101	AC. 44646	2.033	0.033	1.200	0.333	0.002	0.994	0.002	0.002	SP2	RSG, MGR
102	AC. 44604	1.717	0.037	0.622	0.333	0.003	0.951	0.029	0.017	SP2	RSG, MGR
103	AC. 44597	3.583	0.023	1.594	0.333	0.004	0.993	0.002	0.001	SP2	RSG, MGR
104	AC. 44638	3.433	0.030	2.006	0.333	0.001	0.284	0.001	0.714	Admix	RSG, AGR, MGR
105	AC. 44595	2.283	0.023	0.856	0.333	0.003	0.977	0.008	0.012	SP2	RSG, MGR
106	AC. 44588	1.900	0.023	0.717	0.333	0.002	0.994	0.003	0.002	SP2	RSG, MGR
107	AC. 44591	3.100	0.025	1.144	0.333	0.002	0.995	0.003	0.001	SP2	RSG, MGR
108	AC. 44594	5.100	0.045	2.150	0.333	0.006	0.979	0.013	0.002	SP2	RSG, MGR
109	AC. 43737	4.850	0.027	2.617	0.293	0.002	0.991	0.003	0.003	SP2	RSG, AGR, MGR
110	AC. 43660	8.983	0.101	4.450	0.310	0.003	0.993	0.003	0.002	SP2	RSG, AGR, MGR
111	AC. 43732	3.683	0.051	2.089	0.323	0.002	0.995	0.002	0.001	SP2	RSG, AGR, MGR
112	AC. 43661	4.433	0.034	2.356	0.327	0.003	0.988	0.007	0.002	SP2	RSG, AGR, MGR
113	AC. 43738	3.467	0.032	1.556	0.301	0.004	0.991	0.002	0.003	SP2	RSG, AGR, MGR
114	AC. 43669	6.533	0.038	3.722	0.325	0.004	0.987	0.006	0.003	SP2	RSG, AGR, MGR
115	AC. 43663	2.483	0.075	1.689	0.417	0.002	0.994	0.002	0.003	SP2	RSG, AGR, MGR
116	AC. 43658	2.517	0.050	1.633	0.293	0.001	0.996	0.001	0.001	SP2	RSG, AGR, MGR
117	AC. 43662	4.567	0.034	2.250	0.290	0.002	0.96	0.004	0.034	SP2	RSG, AGR, MGR
118	AC. 43670	4.333	0.091	2.294	0.296	0.004	0.816	0.003	0.178	SP2	RSG, AGR, MGR
119	AC. 43675	6.650	0.054	3.828	0.310	0.002	0.98	0.003	0.014	SP2	RSG, AGR, MGR
120	AC. 43676	0.067	0.047	0.756	0.286	0.014	0.935	0.007	0.044	SP2	MGR
121	CR4034-77-56-1	0.692	0.173	0.825	0.189	0.996	0.001	0.002	0.001	SP1	RGR, MGR
122	CR4034-77-25-4	0.725	0.181	0.912	0.178	0.995	0.001	0.002	0.001	SP1	RGR, MGR
123	Jalamagna	0.930	0.114	2.320	0.220	0.996	0.001	0.002	0.001	SP1	AGR, MGR
124	Panidhan	0.900	0.081	1.390	0.201	0.996	0.001	0.002	0.001	SP1	
	CV %	10.56	11.71	12.74	4.17						
	LSD_5%_	0.145	0.051	0.232	0.089						

**Table 2 genes-14-00902-t002:** Estimation of genetic diversity parameters based on 143 SSR marker loci in a panel containing 124 rice germplasm lines.

Sl. No	Marker	No. of Alleles	Major Allele Frequency	Gene Diversity	Heterozygosity	PIC	Inbreeding Coefficient (f)
1	RM5310	4	0.790	0.357	0.032	0.335	0.910
2	RM582	4	0.718	0.454	0.032	0.422	0.930
3	RM13335	4	0.577	0.525	0.008	0.430	0.985
4	RM6275	4	0.730	0.436	0.056	0.402	0.871
5	RM50	4	0.403	0.685	0.024	0.626	0.965
6	RM85	4	0.431	0.669	0.121	0.608	0.821
7	RM222	4	0.625	0.562	0.024	0.524	0.957
8	RM247	5	0.500	0.594	0.065	0.515	0.892
9	RM328	3	0.581	0.570	0.000	0.504	1.000
10	RM337	6	0.431	0.667	0.113	0.609	0.832
11	RM340	5	0.722	0.443	0.097	0.405	0.783
12	RM470	5	0.464	0.689	0.839	0.643	−0.213
13	RM472	3	0.528	0.506	0.089	0.386	0.826
14	RM506	3	0.694	0.450	0.129	0.383	0.715
15	RM1812	3	0.444	0.604	0.000	0.519	1.000
16	RM3701	4	0.677	0.481	0.492	0.425	−0.019
17	RM6947	3	0.871	0.231	0.000	0.215	1.000
18	RM14978	3	0.419	0.637	0.000	0.560	1.000
19	RM18776	3	0.851	0.260	0.024	0.236	0.908
20	RM22034	3	0.919	0.150	0.000	0.143	1.000
21	RM24161	4	0.540	0.613	0.129	0.552	0.791
22	RM223	5	0.649	0.541	0.056	0.509	0.897
23	RM440	5	0.403	0.695	0.266	0.641	0.620
24	RM201	3	0.496	0.574	0.024	0.482	0.958
25	RM216	4	0.528	0.629	0.121	0.574	0.809
26	RM258	3	0.387	0.654	0.000	0.579	1.000
27	RM286	4	0.464	0.631	0.113	0.560	0.823
28	RM3735	4	0.339	0.722	0.960	0.670	−0.325
29	RM1347	3	0.516	0.565	0.000	0.472	1.000
30	RM7571	3	0.706	0.445	0.008	0.388	0.982
31	RM14723	4	0.508	0.634	0.194	0.573	0.697
32	RM103	3	0.492	0.557	0.774	0.458	−0.386
33	RM315	3	0.871	0.228	0.000	0.209	1.000
34	RM225	3	0.508	0.549	0.177	0.448	0.679
35	RM486	3	0.649	0.481	0.105	0.398	0.783
36	RM256	3	0.730	0.403	0.056	0.333	0.861
37	RM1113	3	0.665	0.460	0.056	0.374	0.878
38	RM3423	3	0.484	0.575	0.000	0.483	1.000
39	RM6100	3	0.427	0.644	0.032	0.569	0.950
40	RM590	3	0.718	0.441	0.065	0.395	0.855
41	RM5793	3	0.629	0.531	0.016	0.471	0.970
42	RM405	3	0.685	0.480	0.000	0.432	1.000
43	RM547	5	0.488	0.570	0.161	0.477	0.719
44	RM7364	5	0.633	0.561	0.161	0.529	0.714
45	RM205	3	0.633	0.522	0.024	0.458	0.954
46	RM167	4	0.714	0.452	0.097	0.412	0.787
47	RM229	4	0.363	0.707	0.129	0.652	0.819
48	RM20A	3	0.637	0.523	0.016	0.463	0.969
49	RM235	5	0.399	0.715	0.169	0.667	0.765
50	RM7003	4	0.677	0.491	0.081	0.443	0.837
51	RM5436	4	0.444	0.618	0.056	0.540	0.909
52	RM25181	5	0.399	0.704	0.161	0.653	0.772
53	RM469	3	0.633	0.514	0.040	0.444	0.922
54	RM6547	3	0.871	0.233	0.016	0.220	0.931
55	RM152	4	0.492	0.632	0.016	0.566	0.975
56	RM148	2	0.669	0.443	0.081	0.345	0.819
57	RM421	3	0.476	0.624	0.000	0.549	1.000
58	RM2634	3	0.399	0.655	0.024	0.581	0.963
59	RM248	4	0.335	0.734	0.113	0.685	0.847
60	RM7179	5	0.331	0.763	0.347	0.724	0.548
61	RM215	3	0.597	0.500	0.016	0.397	0.968
62	RM324	4	0.556	0.624	0.153	0.579	0.756
63	RM317	3	0.734	0.395	0.000	0.323	1.000
64	RM174	3	0.524	0.612	0.065	0.543	0.895
65	RM556	3	0.847	0.271	0.032	0.254	0.882
66	RM257	4	0.395	0.662	0.226	0.592	0.661
67	RM502	3	0.815	0.310	0.000	0.275	1.000
68	RM331	4	0.468	0.672	0.056	0.619	0.917
69	RM403	4	0.577	0.581	0.081	0.522	0.862
70	RM309	3	0.698	0.458	0.040	0.403	0.913
71	RM6641	3	0.565	0.585	0.000	0.520	1.000
72	RM3	3	0.371	0.664	0.032	0.590	0.952
73	RM594	3	0.585	0.558	0.008	0.487	0.986
74	RM3392	4	0.488	0.617	0.105	0.545	0.831
75	RM1278	3	0.774	0.372	0.065	0.337	0.828
76	RM168	3	0.629	0.506	0.161	0.428	0.684
77	RM3375	3	0.581	0.567	0.032	0.499	0.944
78	RM282	3	0.734	0.426	0.000	0.387	1.000
79	RM26632	4	0.367	0.701	0.153	0.645	0.783
80	RM1341	3	0.593	0.546	0.024	0.472	0.956
81	RM4112	3	0.504	0.615	0.153	0.542	0.753
82	RM20377	4	0.762	0.382	0.081	0.338	0.790
83	RM210	5	0.367	0.733	0.710	0.685	0.035
84	RM218	4	0.597	0.574	0.032	0.522	0.944
85	RM494	5	0.387	0.712	0.024	0.664	0.966
86	RM336	5	0.387	0.706	0.089	0.655	0.875
87	RM3475	4	0.452	0.662	0.040	0.600	0.940
88	RM480	4	0.552	0.607	0.024	0.551	0.960
89	RM566	4	0.452	0.650	0.016	0.585	0.975
90	RM11701	3	0.653	0.464	0.000	0.370	1.000
91	RM220	6	0.355	0.747	0.194	0.706	0.743
92	RM488	6	0.310	0.753	0.202	0.711	0.734
93	RM6374	6	0.327	0.772	0.073	0.737	0.907
94	RM233	5	0.347	0.728	0.258	0.681	0.648
95	RM112	3	0.879	0.216	0.000	0.199	1.000
96	RM13600	4	0.480	0.663	0.097	0.608	0.855
97	RM495	3	0.613	0.549	0.032	0.490	0.942
98	RM493	7	0.274	0.817	0.573	0.792	0.303
99	RM444	5	0.310	0.776	0.153	0.740	0.804
100	RM468	3	0.778	0.369	0.024	0.338	0.935
101	RM6054	3	0.927	0.137	0.016	0.133	0.883
102	RM509	3	0.766	0.385	0.000	0.351	1.000
103	RM5638	6	0.625	0.574	0.129	0.545	0.777
104	RM8044	6	0.286	0.759	0.226	0.718	0.704
105	RM8271	5	0.415	0.717	0.145	0.673	0.799
106	RM171	4	0.500	0.640	0.056	0.580	0.912
107	RM16686	3	0.403	0.656	0.000	0.582	1.000
108	RM434	4	0.565	0.596	0.024	0.538	0.960
109	RM6091	4	0.823	0.310	0.000	0.291	1.000
110	RM209	4	0.540	0.613	0.000	0.552	1.000
111	RM245	4	0.565	0.593	0.000	0.532	1.000
112	RM1089	4	0.419	0.633	0.065	0.561	0.899
113	RM228	4	0.637	0.533	0.185	0.481	0.654
114	RM401	3	0.762	0.388	0.056	0.351	0.856
115	RM11	3	0.464	0.588	0.008	0.499	0.986
116	RM3351	3	0.597	0.511	0.000	0.415	1.000
117	RM5749	3	0.585	0.505	0.024	0.399	0.952
118	RM335	2	0.730	0.394	0.073	0.317	0.817
119	RM144	3	0.593	0.512	0.169	0.416	0.672
120	RM300	3	0.855	0.258	0.016	0.240	0.938
121	RM1132	4	0.379	0.719	0.032	0.668	0.955
122	RM400	4	0.363	0.717	0.468	0.665	0.352
123	RM471	3	0.790	0.350	0.000	0.318	1.000
124	RM243	3	0.573	0.553	0.016	0.474	0.971
125	RM467	3	0.573	0.566	0.000	0.494	1.000
126	RM564	4	0.452	0.610	0.097	0.530	0.842
127	RM8007	3	0.774	0.375	0.000	0.344	1.000
128	RM441	4	0.476	0.624	0.581	0.553	0.073
129	RM518	3	0.540	0.537	0.000	0.436	1.000
130	RM253	4	0.536	0.602	0.081	0.534	0.867
131	RM274	3	0.677	0.467	0.000	0.399	1.000
132	RM242	4	0.573	0.591	0.016	0.535	0.973
133	RM3231	4	0.343	0.702	0.661	0.643	0.063
134	RM5687	4	0.411	0.693	0.645	0.637	0.073
135	RM5626	3	0.581	0.512	0.742	0.410	−0.446
136	RM452	3	0.460	0.626	0.000	0.548	1.000
137	RM14960	2	0.976	0.047	0.000	0.046	1.000
138	RM558	2	0.968	0.062	0.000	0.060	1.000
139	RM406	2	0.968	0.062	0.000	0.060	1.000
140	RM522	2	0.976	0.047	0.000	0.046	1.000
141	RM10124	2	0.968	0.062	0.000	0.060	1.000
142	RM181	3	0.919	0.151	0.000	0.146	1.000
143	RM175	3	0.887	0.207	0.000	0.197	1.000
	Mean	3.65	0.582	0.530	0.109	0.474	0.796

**Table 3 genes-14-00902-t003:** Analysis of molecular variance (AMOVA) of the panel population for germination rate and early seedling growth parameters in rice.

Sources of Variation	AMOVA for the Four Subpopulations at K = 4			
	df.	Mean Sum of Squares	Estimated Variance	Percentage Variation
Among populations	4	305.33	5.46	14
Among individuals (accessions) within population	119	60.61	26.42	67
Within individuals (accessions)	124	7.77	7.77	20
Total	247		39.65	100
F-Statistics	Value	*p*-value		
F_ST_	0.138			
F_IS_	0.773			
F_IT_	0.804			
F_ST_ max.	0.522			
F’_ST_	0.264			

**Table 4 genes-14-00902-t004:** Marker alleles’ association with germination rate and early seedling growth parameters in the rice germplasm lines present in the panel population detected by both GLM and MLM analyses at *p* < 0.01.

Traits	Marker	Chr #	Position	GLM				MLM			
				Marker_F	Marker_p	*q* Value	Marker_R^2^	F	*p*	*q* Value	Marker_R^2^
RSG	RM337	8	27	27.20863	7.73E−07	7.34E−06	0.0892	16.54241	8.55E−05	0.001159	0.11592
RSG	RM22034	7	56	15.24579	1.56E−04	0.000573	0.05441	7.72005	0.00634	0.007529	0.0541
RSG	RM494	6	221	14.93372	1.81E−04	0.000573	0.05341	14.91786	1.83E−04	0.001159	0.10454
RGR	RM1812	11	44	10.65642	0.00143	0.002236	0.05988	9.21126	0.00295	0.005605	0.07091
AGR	RM337	8	27	16.07491	1.06E−04	0.000504	0.1088	9.56844	0.00246	0.005193	0.07805
AGR	RM7179	6	159	10.79532	0.00133	0.002236	0.07601	10.27457	0.00173	0.005178	0.08381
AGR	RM494	6	221	7.88757	0.00581	0.006494	0.0568	9.96554	0.00202	0.005178	0.08129
AGR	RM16686	4	300	10.75337	0.00136	0.002236	0.07574	11.68419	8.62E−04	0.003276	0.09531
MGR	RM3735	4	80	31.13489	1.52E−07	2.89E−06	0.10146	15.01011	1.75E−04	0.001159	0.1052

## Data Availability

All data generated in this study are included in the article.

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
