# Peer review of "Mapping the Genomic Regions Controlling Germination Rate and Early Seedling Growth Parameters in Rice"

_genes, 2023, doi:10.3390/genes14040902_

Round 1
Reviewer 1 Report
This study investigated four seed vigour traits including the germination rate and three early seedling growth parameters using 278 landrace rice from India. A total of 124 lines were selected and 143 SSR markers were assayed. As a result, eight QTLs were identified using association mapping. The finding would be useful for improvement in rice seed vigour trait.
However, the manuscript was rough in handwriting, eg.
1, All the graphs looked not clear. The resolution seemed to be too low or the word was hard to be distinguished.
2, No tables were presented in the text. The authors should balance tables between text and supplementary data.
3, The format of citation in references was inconsistent. Take examples, “Scientific reports 2019” (line 542); “Agronomy. 2021” (line 548); “Plant Sci” (line 551); “Planta2015” (line 556); the first letter of each word was capitalized (line 573); “32 (Xie et al. 2014)” (line 103), etc.
Some queries and comments:
1, Seed materials. In line 98, the authors introduced that 278 germplasm were collected from the states of Assam, MP, Kerala, Odisha, and Manipur of India. However, this information was missing in the supplementary table 1 and was not further discussed in the text. Furthermore, 124 were shortlisted from 278 lines, but without selection standard in details, just saying they were cut down from all the groups of each trait. Unfortunately, some lines were increased ( Fig 1 to Fig 2). eg. 8 changed to 20 in “Very low” group of RGP? 7 changed to 8 in “High” group of MGR? It could not be happened. In my opinion, it is necessary to check the raw data. My suggestion is to directly use 124 germplam and remove 278 lines in the manuscript.
2, I’m not sure whether the data in Supplementary Table 1 was right. It seemed that the value of mean germination rate (MGR) was much lower than my expectation. Meanwhile, in the text (line110-120), there was no word about how to sore this trait. Instead, germination time was described additionally.
3, The authors stated some markers were associated with seed vigour, For example, RM 337 at 27 cM position (line 413). I can’t find any information in M & M, what “cM” means? How to calculate? Why not use “Mb”? Moreover, the association should be allele rather than marker, herein, allele of “J-1”, rather than other alleles, got the highest association. By the way, the information of “Chr” was missing in supplementary table 3 and the “marker” was inconsistent with ST2.
4, Fig 3B, it is better to present the message in table.
5, Fig 5, it is better to add a trendline in A and unify the words in B and C, eg. marker p and p, marker F and F, R2 and Rsq. Additional, line 319-320, A,B and C was wrong markered.
6, Supplementary table 2, it is better to order the markers by name or Chr and position.
7, Line 137, “30 min” should be “30 sec”. By the way, “s” vs “m”, “sec” vs “min”.
8, Line 185, “274” should be “278”.
9, Line 192-193, these lines could not be checked in Fig 2.
10, Line 208, where A, B and C gone in Fig 2?
11, Lines 366-375 and lines 379-389, did the same meaning were expressed?
Author Response
Comments and Suggestions for Authors
This study investigated four seed vigour traits including the germination rate and three early seedling growth parameters using 278 landrace rice from India. A total of 124 lines were selected and 143 SSR markers were assayed. As a result, eight QTLs were identified using association mapping. The finding would be useful for improvement in rice seed vigour trait.
However, the manuscript was rough in handwriting, eg.
1, All the graphs looked not clear. The resolution seemed to be too low or the word was hard to be distinguished.
Response: We are again submitting higher resolution graphs.
2, No tables were presented in the text. The authors should balance tables between text and supplementary data.
Response: Sorry. We could not detect it. We have now pasted the tables after reference section in the revised manuscript.
3, The format of citation in references was inconsistent. Take examples, “Scientific reports 2019” (line 542); “Agronomy. 2021” (line 548); “Plant Sci” (line 551); “Planta2015” (line 556); the first letter of each word was capitalized (line 573); “32 (Xie et al. 2014)” (line 103), etc.
Response: Thanks. Now we have checked the reference section and corrected all the errors.
Some queries and comments:
1, Seed materials. In line 98, the authors introduced that 278 germplasm were collected from the states of Assam, MP, Kerala, Odisha, and Manipur of India. However, this information was missing in the supplementary table 1 and was not further discussed in the text. Furthermore, 124 were shortlisted from 278 lines, but without selection standard in details, just saying they were cut down from all the groups of each trait. Unfortunately, some lines were increased ( Fig 1 to Fig 2). eg. 8 changed to 20 in “Very low” group of RGP? 7 changed to 8 in “High” group of MGR? It could not be happened. In my opinion, it is necessary to check the raw data. My suggestion is to directly use 124 germplam and remove 278 lines in the manuscript.
Response: W collected 278 germplasm lines from the gene bank and phenotyped the 278 germplasm lines for the 4 studied traits. We developed the panel population containing 124 germplasm lines which is the representative of the original population based on the 4 traits by shortlisting germplasm lines from all the phenotypic groups of each of the 4 studied traits (Table 1). Each parameter is broadly classified into 5 groups based on the groups in the frequency distribution (Fig. 1). This panel is further genotyped and phenotyped for association mapping. As the panel is a representative of 278, so we may use the original number.
2) In Fig. 1 for RGR, there was 8 genotypes in very low group and in Fig. 2 there was 20 genotypes in very low group. Also incase of MGR in Fig.1 there was 7 genotypes in high group and in Fig.2 there was 8 genotypes. The increase in number was due to the inclusion of few genotypes which showed near to the border value of high MGR nd the estimate was based on two replication. The phenotyping of the germplasms of the panel was performed using 3 replications. The estimate in the initial population having border value has therefore showed 7 in initial population to 8 in the final phenotyping for MGR. Incase of RGR, 8 genotypes showed very low in the initial population using 2 replications. However, at higher replications of 3, 20 genotypes showed very low value.
2, I’m not sure whether the data in Supplementary Table 1 was right. It seemed that the value of mean germination rate (MGR) was much lower than my expectation. Meanwhile, in the text (line110-120), there was no word about how to score this trait. Instead, germination time was described additionally.
Response: The value of mean germination rate (MGR) as given in Supplementary Table 1 not low. MGR is not same as Germination value. The mean germination rate is the velocity of germination.It is the inverse of mean germination time (MGT). MGR is expresses as day-1 and ranges from 0 - 1. MGR and MGT was calculated according to the formula of Ellis and Roberts (1981).MGT(day) =SnD/N. Where, n is the number of seeds, germinated on each day, D is number of days counted from the beginning of germination and N is the total number of seeds germinated. MGR(day-1)=1/MGT. Few references are cited for coinfirmation.
Reference:
1.EllisR. A. and E. H. Roberts, The Quantification of Ageing and Survival in Orthodox Seeds,” Seed Science and Technology, 9, 1981, pp. 373-409.
2.Ranal M and De-Santana G D, How and why to measure the germination process, RevistaBrasil. Bot.,2006. 29 (1),1-11. 3.
3.Lazim K S and Ramadhan N M, Mathematical expression study of some germination parameters and the growth by presowing wheat seeds treatment with a static magnetic field and ammonium molybdate. Plant Archives 19 ( 2) 2019 2294-2300.
3, The authors stated some markers were associated with seed vigour, For example, RM 337 at 27 cM position (line 413). I can’t find any information in M & M, what “cM” means? How to calculate? Why not use “Mb”? Moreover, the association should be allele rather than marker, herein, allele of “J-1”, rather than other alleles, got the highest association. By the way, the information of “Chr” was missing in supplementary table 3 and the “marker” was inconsistent with ST2.
Response: We are revising the locations in Mb under results section. But, all the previous workers have reported the studied QTL locations in cM. therefore, we are describing in cM under discussion section.
Thanks for the suggestion allele in place of marker. Infact, that is marker allele and hence we are revising maker allele in place of marker.
In addition, we have now supplied the information of “Chr#” and marker name in supplementary table 3 and “ST4.
4, Fig 3B, it is better to present the message in table.
Response: There are already 4+4 tables. The look of a half matrix table containing only 4 parameters will not be attracting. As number of figures are more, so we have merged it with Fig. 3.
5, Fig 5, it is better to add a trendline in A and unify the words in B and C, eg. Marker p and p, marker F and F, R2 and Rsq. Additional, line 319-320, A,B and C was wrong markered.
Response: Thanks. As this is a software generated figure, we could not unified the words in two figures and hence deleted the figure 5C.
6, Supplementary table 2, it is better to order the markers by name or Chr and position.
Response: We have now arranged the table based on the chromosome#.
7, Line 137, “30 min” should be “30 sec”. By the way, “s” vs “m”, “sec” vs “min”.
Response: Thanks. We have revised in the manuscript.
8, Line 185, “274” should be “278”.
Response: Thanks
9, Line 192-193, these lines could not be checked in Fig 2.
Response. Thanks. We have corrected in place of fig.2 as Fig.7B & Table 1.
10, Line 208, where A, B and C gone in Fig 2?
Response: We have not submitted fig. 2a,b & C. Now, figure legend is corrected as “Frequency distribution of genotypes present in the panel population by histogram for RSG, AGR, MGR and RGR”.
11, Lines 366-375 and lines 379-389, did the same meaning were expressed?
Response: Thanks a lot. Lines 366-375 is repeated again from line 379-389.

Reviewer 2 Report
1. In the Results section: too much description that make the readers not focus and confuse. So, it is needed to revise to become brief and representative description.
2. What is the main question addressed by the research?
ð Yes
3. Do you consider the topic original or relevant in the field? Does it
address a specific gap in the field?
ð I think, the topic is old and general. The research is still using SSR markers which is old markers and not really accurate compared to the new markers, such as SNPs. The measured parameters also just morphological traits, not including physiological traits.
4. What does it add to the subject area compared with other published
material?
ð This research is not really different with previous published material.
5. What specific improvements should the authors consider regarding the
methodology? What further controls should be considered?
ð If it is possible, please use current markers like SNPs not only SSR markers.
6. Are the conclusions consistent with the evidence and arguments presented
and do they address the main question posed?
ð Yes, but still need more focus to address the main question.
7. Are the references appropriate?
ð Yes, the references are appropriate.
8. Please include any additional comments on the tables and figures.
ð Please increase the resolution for all the figures because the figures look blurry.
ð Please repair the layout for all the figures.
ð Please choose the figures that really relevant to the results.
Author Response
Comments and Suggestions for Authors
Query 1: In the Results section: too much description that make the readers not focus and confuse. So, it is needed to revise to become brief and representative description.
Response: Yes, we have revised the results section and prepared a brief description.
Query 2: What is the main question addressed by the research?
Response: Seed vigour is an important trait of good quality seed that ensures germination, seedling growth, and establishment of seedling in the field and withstand adverse climatic conditions. Additionally, seed vigour is very important for the direct seeded rice which enhances crop establishment and increases weed competitiveness. Seed vigour improvement is a breeding challenge in rice research. Therefore, transfer of seedling vigour and its related parameters into high yielding varieties of rainfed rice is very important. We have introduced the topic under introduction section.
Query 3: Do you consider the topic original or relevant in the field? Does it
address a specific gap in the field? I think, the topic is old and general. The research is still using SSR markers which is old markers and not really accurate compared to the new markers, such as SNPs. The measured parameters also just morphological traits, not including physiological traits.
Response: Germination rate and early seedling growth parameters in rice ensure germination, seedling growth, and establishment of seedling in the field and withstand adverse climatic conditions. Early seedling growth parameters are also very important for the direct seeded rice which enhances crop establishment and increases weed competitiveness. Control of these traits is not well studied. Chromosomal location of the QTL controlling of these traits are reported very few only. Seed vigour improvement is a breeding challenge in rice research.
In this study, we have used higher number of SSR markers i.e.143 in number. SSR markers will differentiate the heterozygote and homozygote individual for the target gene/QTL in the screening of progenies through foreground selection. Genetic analysis using simple sequence repeat markers are useful for their hyper variable, co-dominant, multi-allelic, robust and chromosome specific in nature. Hence, very much helpful and accurate in prediction in screening of breeding population.
The 4 studied early seedling growth parameters viz., rate of shoot growth (RSG), absolute growth rate (AGR), mean germination rate (MGR) and relative growth rate (RGR) are physiological parameters.
Query 4: What does it add to the subject area compared with other published
material? ð This research is not really different with previous published material.
Response: The results published earlier on genetic analyses of seedling growth parameters were mostly using the derived population from cross of two parents while very few were by association mapping.
Query 5: What specific improvements should the authors consider regarding the
methodology? What further controls should be considered?
ð If it is possible, please use current markers like SNPs not only SSR markers.
Response: We do not have SNP genotyping platform. Therefore, we have used SSR markers.
Query 6: Are the conclusions consistent with the evidence and arguments presented
and do they address the main question posed?
ð Yes, but still need more focus to address the main question.
Response: Thanks
Query 7: Are the references appropriate?
ð Yes, the references are appropriate.
Response: Thanks
Query 8: Please include any additional comments on the tables and figures.
ð Please increase the resolution for all the figures because the figures look blurry.
ð Please repair the layout for all the figures.
ð Please choose the figures that really relevant to the results.
Response: We have increased the resolution of all the figures in the revised submission. We have removed the figure 5c.

Reviewer 3 Report
This paper entitled “Mapping for the genomic regions controlling germination rate and early seedling growth parameters in rice” identified 8 novel QTLs controlling the vigorous seedling traits and might be helpful to for rice breeding programs.
1. L117, please reduce the font size “the time for the seed to germination…”
2. Please improve the resolution of Fig. 7 and Fig. 8.
3. L366, change phynotypic to “phenotypic”
4. Suggest the authors can place the Fig.9(C) the positions of the QTLs on the chromosomes for RSG, RGR, AGR and MGR detected by association mapping in rice, in a separate Figure and also add the known markers in the map for compariso.
5. L411, High R2 value of >.1, change to “>0.1”.
6. L446, AC. 9058 and AC. 9063 change to “AC. 9058, and AC. 9063”; L464 “RM16686, and RM3735”, and others in the text.
7. L474, Change Wang et al…[12] to “Wang et al. [12]”
Author Response
Comments and Suggestions for Authors
This paper entitled “Mapping for the genomic regions controlling germination rate and early seedling growth parameters in rice” identified 8 novel QTLs controlling the vigorous seedling traits and might be helpful to for rice breeding programs.
Query 1: L117, please reduce the font size “the time for the seed to germination…”
Response: Thanks. We have reduced the font size of “the time for the seed to germination…”
Query 2: Please improve the resolution of Fig. 7 and Fig. 8.
Response: We have now increased the esolution all the figures.
Query 3: L366, change phynotypic to “phenotypic”
Response: thanks. We have changed phynotypic to “phenotypic”.
Query 4: Suggest the authors can place the Fig.9(C) the positions of the QTLs on the chromosomes for RSG, RGR, AGR and MGR detected by association mapping in rice, in a separate Figure and also add the known markers in the map for comparison.
Response: The positions of the QTLs on the chromosomes for RSG, RGR, AGR and MGR detected are shown along with other known markers in the revised submission.
Query 5. L411, High R2 value of >.1, change to “>0.1”.
Response: We have changed to “>0.1”.
Query 6. L446, AC. 9058 and AC. 9063 change to “AC. 9058, and AC. 9063”; L464 “RM16686, and RM3735”, and others in the text.
Response: Thanks. We have changed in L446, AC. 9058 and AC. 9063 change to “AC. 9058, and AC. 9063”; L464 “RM16686, and RM3735”, and others in the text.
- L474, Change Wang et al…[12] to “Wang et al. [12]”
Response: Thanks. We have changed in Wang et al..[12] to “Wang et al. [12]”

Round 2
Reviewer 1 Report
I’m afraid that the revised figures can not meet the requirement of publication so far.
Others not changed:
1, “ (Xie et al. 2014)” (line 103) still be there.
2, Table 2 and supplementary table 2, the markers were not arranged based on the chromosome#.
Reviewer 2 Report
Great improvement.